# An Evaluation of the United Kingdom Motor Neuron Disease Nurses and Allied Health Professionals (UK MND NAHP) Workforce: A Census

Isaac Chau®⬚*, Ethan Stoker®⬚, Jessica Gill®⬚, Judith Newton®⬚, on behalf of The United Kingdom Motor Neuron Disease Nurses and Allied Health Professionals Consortium¶

Anne Rowling Regenerative Neurology Clinic, Centre for Clinical Brain Sciences, University of Edinburgh, Edinburgh, Scotland, United Kingdom

¶ Membership of The United Kingdom Motor Neuron Disease Nurses and Allied Health Professionals Consortium is provided in the Acknowledgements.
® These authors contributed equally to this work.
¤ Current address: Chancellor's Building, University of Edinburgh, Edinburgh BioQuarter, 49 Little France Crescent, Edinburgh, EH16 4SB
* ichau@ed.ac.uk

## Abstract

According to the National Institute for Health and Care Excellence, motor neuron disease assessment and management should be a coordinated, clinic-based, multidisciplinary team approach. However, the wellbeing, work experiences, and the alignment with national frameworks and standards of the motor neuron disease (MND) nurses and allied health professionals' workforce are severely underreported and under-researched within the literature. Therefore, this report aimed to capture the workforce and their alignment with national frameworks and standards, and to assess their experiences working as an MND health care professional. A pragmatic research paradigm and a mixed methods approach was employed using a cross-sectional questionnaire survey to collect, compare, and interpret quantitative and qualitative data points. Data was gathered under the remit of an audit and service evaluation under NHS Lothian. Demographics data and work-related characteristics were collected. Job experience and wellbeing were collected using *Likert* scales and open-ended questions. The level of burnout was assessed using the validated tool, the Burnout Assessment Tool (BAT). Compliance with national frameworks were based on the NICE guidelines and the Scottish MND Advanced Clinical Nurse Specialist Pillars of Practice Competencies. 64 HCPs completed the questionnaire, with the majority of respondents from England (54.7%) and Scotland (35.9%). Education level was mainly having a Bachelors (or equivalent) degree (40%) or a Masters (or equivalent) degree (31%), with the remaining having a diploma-based qualification (29%). The analysis revealed three key themes: the importance of the multi-disciplinary team

**Data availability statement:** Data are available at https://doi.org/10.7488/ds/7746

**Funding:** The author(s) received no specific funding for this work.

**Competing interests:** The authors have declared that no competing interests exist.

(MDT), the roles and the level of competencies, and the benefits and challenges in providing direct care. This was associated with 14% and 12% of respondents being at medium and high risk of burnout, respectively. This report highlighted the importance of a collaborative MDT to support the needs of patients, their carers/ family members and HCPs themselves. The workforce found the flexibility, autonomy, and variety within their role beneficial where almost 80% of the respondents engaged in 8 of the 15 competencies. The benefits of providing direct care were found to be associated with feeling valued by the patients, their carers/ family members, and the core and extended MDT, and feeling satisfied about their work. The perceived challenges of providing direct care involved isolation, lack of direct funding, and a high caseload with complex needs and not enough time to provide quality care. It is recommended that a national competency programme or a Masters level course in MND care should be developed to maintain the quality of care, and future research should aim to evaluate the entire workforce longitudinally, address organisational barriers, and explore burnout preventative strategies to maintain a resilient workforce.

## Introduction

### Background

Motor neuron disease (MND) is a group of neurodegenerative disorders that leads to premature death 2–4 years after diagnosis, often caused by respiratory failure due to the progressive degeneration of either or both upper or lower motor neurons [1]. There is currently no cure and the single disease-modifying treatment licensed for use in the United Kingdom (UK), Riluzole, only reduces the mortality rate by 38.6% [2] and prolongs survival by another 2–3 months [3]. As a diagnosis may take up to 12–14 months [4], patients are frequently already at an advanced stage of disease upon diagnosis, and this can have profound effects on the patient, carers/ family members, and also the multidisciplinary care team [5]. According to the National Institute for Health and Care Excellence (NICE), it is recommended that the organisation of care for MND assessment and management should include a coordinated, clinic-based, multidisciplinary team (MDT) approach [6]. This aligns with the guidelines and recommendations of the European Federation of Neurological Societies (EFNS) which state that attendance by the MDT should be available for patients and communication between the specialist teams should be effective and coordinated [7]. The core MDT should include a neurologist consultant, specialist nurse, dietician, physiotherapist, occupational therapist, respiratory physiologist or a healthcare professional (HCP) who can assess respiratory function, speech and language therapist, and an HCP who specialises in palliative care [6]. Much of the literature on the MDT has been focused on the complexities of delivering care [5,8,9] and their role in managing MND [10,11].

On the other hand, a scoping review [12] identified a severe lack of research on the wellbeing of HCPs who provide care to people with MND (pwMND), although

stress, emotional burden, psychological burden, and burnout are commonly experienced in HCPs involved in progressive neurological diseases [13]. The majority of the literature related to mental health and wellbeing are studies to do with pwMND [14], family members [15], and their caregivers [16,17]. Additionally, more research was conducted on the psychological interventions to cope with MND [12,18–23]. Hence, there is a lack of the aforementioned knowledge on the nurses and allied health professionals (NAHPs) who serve those living with MND in the UK, and importantly their experiences and wellbeing as HCPs who provide MND care. Furthermore, the compliance with national frameworks and standards by each speciality of the MDT across the different NHS localities and health boards has not been formally audited and the understanding of this is unclear.

A census on the NAHPs in the UK who currently serves pwMND has never been done before, and their experiences and wellbeing have not been formally reported. Furthermore, it was reported that pwMND were able to cope with MND because of the emotional support and trust that HCPs provide [14], hence, the gap to understand the experiences and emotional wellbeing of the NAHPs that comprise the MND workforce should be addressed. It was imperative that a holistic representation of the workforce could be gathered to highlight areas of challenges and barriers that could impact their adherence to national frameworks, as well as their experiences and wellbeing, and this could potentially inform policy change and guidance to increase support for the workforce.

### Aim and research questions

Across the UK, several healthcare governing bodies carried out a census to understand the size of their workforce by various demographics such as their job grading, number of working hours, and geographical spread [24]. The Royal College of Paediatrics and Child Health (RCPCH) had published a workforce census in 2019 and found a lack of training positions in Scotland, and this provided enough evidence to support an increase in training positions [25]. The findings from their 2022 census also provided key recommendations that advocated National Health Service (NHS) organisations and leaders to pursue support for the wellbeing of the paediatric and child health workforce, and to ensure equality, diversity, and inclusion needs are met [26].

The aim of this report was to capture the UK MND NAHP workforce and their alignment with national frameworks and standards, and to assess their experiences working as MND HCPs. This was achieved by producing a census questionnaire that was distributed on paper during the inaugural UK MND NAHP meeting in Edinburgh on November 16th, 2023, with more than 80 attendees across the UK including MND clinical nurse specialists, AHP, and MND research teams.

This report aimed to address the following research questions:

1. What is the size (by number) of the MND NAHP workforce in the UK?

2. What is the proportion of UK MND services that comply with national frameworks and standards?

3. What are the experiences and wellbeing of the UK MND NAHP workforce, based on the Burnout Assessment Test (BAT-C and BAT-S)?

4. What are the themes that could be identified in relation to the NHS locality, demographics, and the mental health and wellbeing among the UK MND NAHP workforce?

Therefore, this report hypothesises that:

1. The UK MND NAHP workforce is mainly made up of clinical and experienced HCPs

2. Alignment with national frameworks and standards is adhered to by a high proportion of the workforce

3. There is a high proportion of burnout within the UK MND NAHP workforce based on the Burnout Assessment Tests (BAT-C and BAT-S) due to the terminal nature of MND

## Methods

### Study design

The research paradigm for this census was pragmatism, with a flexible and undefined ontology and epistemology, and offered a practical viewpoint to answer the aims and research questions previously defined [27]. The pragmatic research paradigm was also utilised for the unique setting of a census as this meeting was the first of its kind in the UK to invite all the NAHPs that work within the capacity of MND care; to take advantage of an opportunity to develop the first UK MND NAHP census. The aims of the census were presented at the inaugural UK MND NAHP meeting and the attendees were asked to participate in the census by completing a survey questionnaire. The survey questionnaire was distributed in the inaugural UK MND NAHP meeting on November 16th 2023 in paper form and the online form was distributed on November 17th 2023. The online form was open for one week and closed on November 24th 2023. This meeting covered a variety of topics related to workforce and organisation development, and discussed the future of clinical care within MND services. A variety of NAHPs involved in clinical care, research, and managerial and organisational roles from Scotland, England, Wales, and Northern Ireland were represented in-person and online.

The study design employed a convergent mixed methods approach, using a cross-sectional questionnaire survey with quantitative and qualitative data points that were collated and compared and then interpreted [27]. The mixed methods were used to suit the exploratory aims and make use of the unique setting to gather as much data as possible [28].

### Ethical considerations

The survey questionnaire was distributed in paper form on November 16th 2023 during the UK MND NAHP meeting and the online format was distributed through email on November 17th 2023. The online form was open for a week and closed on November 24th 2023. Consent was given implicitly, voluntarily, and independent of attendance to the meeting. Verbal or written consent was waived and not obtained as this was under the remit of an audit and service evaluation. No personal data was collected and the results from the questionnaire was kept within a secure and encrypted electronic database for at least twelve months while the results from the study will be kept for at least five years. Paper copies of the questionnaire will be stored in a secure location on site of the Anne Rowling Regenerative Neurology Clinic. These will be inputted onto the electronic system by Ethan Stoker.

### Ethics statement

This data was gathered under the remit of an audit and service evaluation under National Health Service (NHS) health board Lothian. As the census did not gather patient identifiable information and had the main aim of assessing the standard of care, verbal and written consent was waived and it was deemed that the aim and design of the census fit within the definition of an audit and service evaluation within NHS Lothian Quality Improvement guidelines. Approval was granted by NHS Lothian on 3rd November 2023 by email confirmation from the administrator of the quality directorate on behalf of NHS Lothian.

### Sampling strategy

Participants were recruited through a convenience approach through an online/paper questionnaire. Participation was voluntary and the data was anonymised, collected, transcribed and stored on the Jisc Online Surveys platform before transferring to local secure servers where it will be held for a year. Study eligibility was based on participants self-reporting their occupation.

### Data collection

The questionnaire and the questions were formulated by an expert panel that included a MND nurse consultant, MND research nurses, and a MND trial assistant, and guided by the national frameworks to ensure the exploratory aims of

the census were met and to make use of the unique opportunity. The questionnaire was piloted by three clinical and two non-clinical members of staff from the Anne Rowling Regenerative Neurology clinic as they were not part of the authorship.

Demographics data and work-related characteristics were collected in the census and used to consistently measure the remaining quantitative variables. These data points were also used to create a sample that can be used to assess the size of the MND NAHPs workforce.

To assess general compliance with national frameworks, questions were formulated from the NICE guidelines [6] and the Scottish MND Advanced Clinical Nurse Specialist Pillars of Practice Competencies [29] to formulate data on the scope of practice in general and the availability to provide best practice as per NICE guidelines. This included using multiple choice questions where participants would select the relevant aspect of care with which they were involved on a regular basis, along with direct questions on caseload and vacancies within the MDT.

To assess the experiences and wellbeing of the MND NAHP workforce, questions were asked about satisfaction and feelings of being valued. Questions of satisfaction and feelings of being valued were assessed through a 4-point Likert scale, ranging from 1 (Never) and up to 4 (Always). These values were used for analysis to assess the satisfaction and the feeling of being valued across a range of facets seen within the MND service and were deemed suitable to assess experiences of NAHPs.

In addition to the satisfaction and valuation scores, the wellbeing of the MND NAHP workforce was assessed by measuring burnout, through the use of a validated assessment tool called the Burnout Assessment Tool-Core dimensions (BAT-C) and Secondary dimensions (BAT-S) developed by Schaufeli, De Witte, and Desart [30]. The BAT-C and BAT-S are self-reported questionnaires that aims to assess burnout. The BAT-C comprises four core dimensions that assess exhaustion, mental distance, impaired emotional, and cognitive control; the BAT-S consists of two secondary dimensions that assess psychological and psychosomatic complaints [30]. To measure the degree of burnout, the BAT-C contains a 33-item set of questions that used a 5-point Likert scale, ranging from 1 (Never) to 5 (Always). The scores were averaged across five subscales in addition to an overall average that was formulated. These scores were then compared with the cut-off values for employees located in the User Manual [30] that allowed the degree and risk of burnout to be categorised as: 1) Green (No Burnout) where the score is between 1.00 to 2.58, 2) Amber (At-Risk of Burnout) where the score is between 2.59 to 3.01, 3) Red (High Risk of Burnout) where the score is between 3.02 to 5.00. The BAT-C and BAT-S assessment was found to be reliable and to have factorial and construct validity [30], and was deemed suitable for the aims of the census. The full assessment tool can be seen in the Tables in S7A and S7B Table.

For the qualitative aspect of the census, open ended questions were formulated at the end of the census to allow the participants a platform to comment on the positive aspects of their job, along with some of the issues and limitations that the participants could identify.

## Data analysis

For the quantitative analysis, the data was cleaned and any abnormalities identified were addressed before analysis. This included highlighting missing data points, along with coding specific answers, grouping demographic categories (e.g job role), and equalising units for the continuous data sets. Missing data points were removed from the specific tests however the remaining data was used for the other points of analysis. Descriptive statistics were used to summarise all variables. Chi-square test were used to analyse for associations between categorical variables such as participant demographics, core MDT vacancies, engagement in the aspects of care, healthcare professionals involved with the core MDT, and established relationships with services. Ordinal variables, including clinical satisfaction scores, were analysed using non-parametric tests. Continuous variables, such as the time worked in MND care, hours worked per week, total number of patients in caseload, and the total score from the Burnout Assessment Tool, were analysed using one-way ANOVA, provided assumptions of normality and equal variances were met. Data analysis was conducted in R. For the

work experience and wellbeing analysis, only data from the clinical professionals was used in order to ensure validity and relevance of the themes developed, and therefore the responses from non-clinical professionals and who are not a nurse or allied health professional (i.e. doctors) were excluded from the analysis.

For the qualitative analysis, a basic content analysis was employed; an inductive and descriptive qualitative approach that aims to develop a mid-range theory from text-based data [31]. Using a pragmatic research paradigm, the free text answers were seen to analyse part of the constructed viewpoints of the participants [27]. This analysis made use of an inductive approach, due to the lack of any suitable theoretical framework. A conventional content analysis was used to answer the descriptive and exploratory aims of the research [31]. For this purpose, a qualitative researcher attempted immersion within the data, making notes and memos for initial thoughts and concepts. After this, the initial set of in-vivo codes and annotations was produced with the aim of summarising the data in a descriptive manner. A second researcher developed a separate coding scheme that was used for comparison to allow further discussion and revisions of the coding scheme and to improve the reliability. These initial codes were then further categorised into broader groups, where links and relationships were examined to formulate themes. These themes were then triangulated with the quantitative data to extend the validity of the results. Data was analysed using NVivo 14 [32], a data management and analysis software.

## Results

### Descriptive statistics

Initial inferential analyses using one-way ANOVA for continuous variables and Chi-square tests for categorical variables revealed no statistically significant differences between groups across the main variables of interest. Given the lack of significant associations, and in order to more effectively explore and illustrate patterns within the data, the authors opted to present the findings primarily using descriptive statistics. This approach allowed for a clearer representation of trends and distributions across participant demographics, engagement in care, and professional experiences within the MDT.

Table in S1 Table shows the demographic distribution of the NAHPs who completed the census. The 64 who completed the census questionnaire represented 75% of the eligible attendees at the nationwide meeting. The Table in S2 Table, the geographic location of the respondents was mostly between England (54.7%) and Scotland (35.9%). Those with a Masters (or equivalent) degree accounted for 31% of the sample, with the majority (40%) having a Bachelors (or equivalent) degree, and the remaining respondents having a diploma-based qualification (29%). The split between working full-time and part-time hours was almost even, with 54.7% of respondents working full-time hours. Those who said they worked part time had hours that fluctuated between 15–30 hours a week. Most of the respondents (n = 35) classified themselves as specialist nurses, followed by care-coordinators (n = 15), research nurses (n = 10), and other AHPs (n = 4).

When viewing the demographic data for the clinical professionals (which includes the specialists nurses and AHPs), 74.4% of the respondents were Band 7, 17.9% of participants were in Band 6 and the remaining participants (7.7%) were in Band 8. Furthermore, previous experience within MND-related care, and time spent in the current post varied significantly, signifying a diverse workforce with a mixture of expertise and experience. For the clinical professionals, the number of patients on the caseload was highly variable, with a median number of 45. The full set of demographic data can be seen in Tables in S1 Table, S2 Table, and S3 Table.

### Theme one: The multidisciplinary team

The first theme to be developed from the data is the importance of the MDT. This was a prominent and recurring theme within the open textbox questions, where many respondents discussed the importance of the MDT to meet the needs of patients, their families, and the health professionals themselves.

*Having a core MDT is essential as we have 'on hand' expert advice and information for patients, families and other health care professionals.* (Respondent 118421608)

*MDT working is fantastic – swift access to support, excellent peer support. Collaborating with regional teams and our community teams such as DNs, care agencies etc.* (Respondent 118972185)

*\*Note: DNs =District Nurses*

These positive responses can be correlated with the Table in S4B Table where, out of the respondents who were clinical professionals, a vast majority reported working with various members of the 'core MDT' as recommended in the NICE guidelines [6], with particularly high rates seen among those who work with neurologists, clinical specialist nurses, and dieticians. Furthermore, over 80% of clinical professionals reported relationships with respiratory services, gastroenterology, wheelchair services, and assistive technology services (as seen in the Table in S4C Table), which adheres to the NICE [6] recommendations for MND services.

However, core MDT members such as palliative specialists, and extended services such as clinical psychology and counselling was identified as having significantly low rates. Furthermore, almost 30% of clinical professionals reported vacancies within the core MDT.

The issues of these vacancies and limitations within the MDT was identified within the content analysis. There was a plethora of responses that identified vacancies as a limitation and issue that affected patient care and produced a "knock-on" effect, as highlighted by this respondent

*We as a team run short staffed which has a knock-on effect* (Respondent 118442902)

Furthermore, many respondents identified how issues in accessing the extended services caused delays and hardships for pwMND and their carers while also producing a negative effect on the staff themselves.

*Access to other consultant specialities, i.e., ENT or maxillofacial* (Respondent 118420947)

*No psychology and limited access to counselling services* (Respondent 118713457)

*\*Note: ENT= Ear, Nose, and Throat*

**Theme two: Roles and responsibilities**

The second theme developed from the data refers to the relevant competencies and conceptualisations of the MND NAHP workforce. As seen in the Table in S4A Table, the clinical professionals were associated with a range of competencies as detailed in the Pillars of Practice Competency Framework [29] for the Scottish MND nurses. This can form links with some of the qualitative codes and sub-themes on the perceived benefits of the flexibility, autonomy, and variety seen within the clinical care. Respondents felt that they had the freedom to manage their own time, in terms of providing care and developing competencies in a challenging and evolving work environment.

*Flexibility of being able to manage own time and prioritise, in particularly assigning sufficient time to meet patient needs at clinic appointments or home visits* (Respondent 118709878)

*varied, enjoy the mixture of job roles challenging, learn new things all the time* (Respondent 118455679)

In addition to the flexibility of the role, the results from the Table in S4A Table also indicated a shared idea of "key competencies" and expertise possessed by clinical professionals involved in MND care. For example, almost 97% of clinical professionals engage in the provision of information and expertise, while up to 90% of clinical professionals engage in

advocacy for the patient and their family, anticipatory care planning, leadership and collaboration with the MDT, the facilitation of learning, and palliative care. This can then be converged with the results from the qualitative data where these competencies on the provision of information, advocacy, and communication were highlighted by respondents.

*Joining the dots as one family described my role to ensure patient centred coordinated care.*

(Respondent 118420738)

*communication with patients, their families, partners and carers; collaboration between*

*specialists and opportunities for professional development* (Respondent 118458816)

*Being able to help advise and signpost to services.* (Respondent 118540331)

However, this flexibility and variability in competencies was not always positive. From the qualitative themes, it was recognised how the often-nebulous definitions of what the MND service is, and what the related health professionals provide, caused conflict when it did not align with patients' and carers/ family members' expectations. This was then expanded to the rest of the MDT, as they may not specialise in MND care, were sometimes not aware of the full complexity of the condition or what the MND services offers.

*We are not here 24/7 and we are not an emergency service – expectations of patients, families and colleagues are often exceptionally high. Colleagues (outside the MDT) often have high and unrealistic expectations of the support we can and should provide.* (Respondent 118972185)

Expanding on this through the Table in S4A Table it can be seen that some of the competencies that were identified in the Pillars of Practice [29] had varied engagement from the respondents, where competencies such as Respiratory Assessment, Specialist Medication Management, and Muscle Management had only 50% recorded engagement from clinical professionals.

### Theme three: Benefits and challenges in providing direct care

The final theme developed from the data is the overall perceived benefits and challenges in providing care to pwMND. The most prominent aspect of the data collected was the benefits of direct care and engagement with the patients and their carers/ family members. As seen in the Table in S5 Table, 100% of participants felt often or always valued by patients and their families, which was further discussed in the qualitative data where respondents expressed gratitude to the individuals they look after, alongside their families and carers, and highlighted this as a key benefit in their current post.

*Being able to support patients, their families, carers at a very difficult and challenging time of*

*their life.* (Respondent 118923353)

*Working with such a lovely cohort of people at such a difficult time in their lives* (Respondent 118709722)

Furthermore, this provision of direct care allowed for close relationships with the MDT and extended services, where 95% of respondents feeling always or often valued by the 'core MDT' and 88% of respondents feeling always or often valued by the 'extended' MDT. This can then be linked with the previous theme on the MDT where the working relationships remain an essential part of the service. However, it is worth mentioning that some of the respondents did not often feel valued by the MDT, and as highlighted in the previous theme, this can cause limitations in care.

*I feel like I'm part of a much wider team to make a difference in MND* (Respondent 118736747)

Another recurring sub-theme on the benefits of providing direct care was the job satisfaction. Many respondents described the benefits of seeing the impact of the care they provide and found the ability to create change and improve the service a rewarding aspect of the job.

*If I can do one thing to make a difference in my patients life than I have done my job*

(Respondent 118713352)

This is seen in the quantitative data (see the Table in S5 Table) where respondents felt largely satisfied in most areas of their job. This includes 100% of respondents feeling always or often satisfied in the provision of information and expertise, 98% of respondents feeling always or often satisfied in their clinical specialities, 93% of respondents feeling always or often satisfied in the provision of collaboration and leadership within the MDT, and 88% of respondents feeling always or often satisfied in the facilitation of learning and education. However, linking back into the theme of roles and responsibilities and specifically the sub-theme on the issue of competencies, up to 35% of respondents responded that they sometimes or never felt satisfied in the engagement with research.

The largely positive responses from the respondents were further seen in the analysis of the BAT-C scores. It was identified that 82% of respondents were at a low risk of burnout across the various sub-sections (see the Table in S6A Table). However, when breaking down the total score into the individual sub-sections, it can be identified how there exists issues that correspond with the qualitative sub-themes describing the negatives of providing direct care.

For example, when assessing the exhaustion sub-section of the BAT-C, it can be seen that 14% of respondents were identified to be at medium risk of burnout and 12% of participants were at high risk of burnout. When linking with the qualitative data, comments on isolation, rurality, and issues with lack of time and high demand (such as with the high caseload numbers, which were shown to vary in the Table in S6A Table) were all associated with feelings of exhaustion and not being able to provide the care they wanted to be able to provide.

*Very time limited, currently not enough time to provide a gold standard service* (Respondent 118709818).

Expanding on from this, when assessing the emotional impairment and cognitive impairment sub-sections of the BAT-C, the results indicate that 24% and 21% were at a medium and high risk of burnout, respectively. This links into some of the pressures of the job as reported by respondents within the qualitative data. The sub-theme of organisational barriers was identified, with frequently reported issues ranging from a lack of direct funding to additional resources such as room space, social care services, and home adaptions. Furthermore, organisational barriers such as poor communication between NHS Trusts was highlighted, linking into the nebulous definition of job responsibilities, and the issue some respondents have when working across NHS Trusts and different community systems.

*The financial burden of MND/ care and support/ adaptions on people living in England.*

*Availability of social care packages. Numerous electronic record systems across the wider*

*MDT across trusts/ social care systems which are not viewable unless you are employed*

*within the applicable trust* (Respondent 118910045).

Finally, it was also discussed by respondents, how the nature of the work can be challenging as the patient group is complex and often has a variety of care needs at varying intensities. This created challenges and restrictions to the care provided to pwMND that was often exacerbated by the organisational factors and increased demand.

*Conflicting priorities so unable to focus all the services/ patient needs.* (Respondent 118758297).

## Discussion

This was the first ever census conducted to evaluate the MND NAHP workforce in the UK using a mixed-methods approach. A mixed-methods approach was utilised because it generates both quantitative and qualitative data, providing more depth and a holistic understanding of the results [33]. This report aimed to understand the size of the workforce, alignment to national frameworks and standards, and identify their experiences and wellbeing working in this area. The analysis revealed that the workforce mainly consists of clinical band 7 nurses who have worked an average of 9 years in delivering MND care; this demonstrates a relatively experienced workforce. The results from 64 respondents revealed three key themes: the importance of the MDT, the roles and the level of competencies, and the benefits and challenges in providing direct care. While these findings generated new insights in this area of MND care, they also highlighted the need for further exploration if these findings are generalisable to the wider MND workforce.

### Theme one: The multidisciplinary team

The results showed that the UK MND NAHP workforce values the importance of having a core MDT; many of the respondents commented that the collaboration with other specialities was important to support the needs of patients, their carers/family members, and themselves. The main function of an MDT is to foster cooperation among HCPs from different fields to determine a patient's treatment plan [34]. A systematic review and meta-analysis evaluated the effectiveness of MDT care on survival and the quality of life of pwMND and found an improvement in survival compared to general neurology care [35]. This was especially prominent in patients with bulbar onset, but showed no overall benefit in health-related quality of life. More specifically, it was found that management by an MDT in England was associated with a survival time of nine months longer than average [36]. A study of the South East London found that pwMND who were supported by an MDT survived 2.5 months longer than those who did not have an integrated MDT within their health board [37], and studies from the Republic of Ireland and Northern Ireland found similar results [38,39].

The results showed that a high percentage of respondents have established relationships with other specialities and also advocate and link their patients and their family members with the wider MDT. More than 70% of the respondents reported that their core MDT included a neurologist, clinical nurse specialist, dietician, physiotherapist, respiratory specialist, and speech and language therapist, while more than 80% reported established relationships with respiratory ventilation services, gastroenterology, wheelchair services, and assistive technology services. This multidisciplinary approach in MND care aligns with the NICE [6] recommendations for MND services. The *Organisation of Care (1.5)* recommendation states that MND care should be provided in a coordinated, clinic-based, specialist MND multidisciplinary approach (1.5.1) with the inclusion of (1.5.4) and an established relationship with (1.5.5) HCPs across different specialties. There are other recommendations that may require further evaluation beyond the scope of our analysis, mainly regarding understanding the frequency (1.5.6), arrangement (1.5.7), and format (1.5.8) of MDT assessments. The NICE [6] guidelines also states that patients are to have access to palliative care (1.5.11); conversely, our results showed that 58.97% of MDTs have a palliative specialist, with an additional 30% of respondents reporting vacancies within their MDT. Comments from the respondents expanded on the effects of this finding and they reported that a lack of communication and long wait times from social, palliative, and psychological services, in addition to clinical care teams being short-staffed have caused limitations in patient care and this had a "knock-on" effect on their patient care. Similar findings were found in a study in Northwest England in which a number of patients and carers were either never referred to social work or that referral only occurred in the end-stage of the disease [40]. Furthermore, a qualitative systematic review that explored the experiences and the need for palliative care in pwMND found that palliative care discussions did not occur explicitly until the end-of-life stage of the disease, and this appeared to be the only time-point when the discussion occurred [41].

The findings from this report suggest that there has been a lack of specialist palliative, psychological, and social care involvement and this may affect the decision-making and timing of implementing life-extending or symptomatic measures, and ultimately the overall health and quality of life of patients, and their carers/ family members [5]. Patients and their

carers value their involvement with the MDT [40] and HCPs who were caring and sensitive to their needs were found to be important to patients because it helps them cope, especially patients in palliative care [14]. To better manage such a heterogenous and progressive disease, future exploration on these vacancies and the lack of involvement of these services is recommended to inform funding and strengthen the capacity of the MDT.

## Theme two: Roles and responsibilities

The results showed that the UK MND NAHP workforce found the flexibility, autonomy, and variety within their role beneficial. Additionally, based on the Pillars of Practice Competency Framework [29] that was produced for the Scottish MND clinical nurse specialists, almost 80% of the respondents engaged in 8 of the 15 competencies.

Currently, there is not a standardised competency framework for the clinical NAHP who provide MND care in the UK. On the contrary, it was found that there are 8 different general guidance documents' for MND care produced by a range of organisations and health boards in the UK. This includes NICE [6] and the National Neurosciences Advisory Group [36], and several health boards in England including the multidisciplinary Pan Dorset MND group [42], the Middlesbrough MND care centre [43], the Greater Manchester MND care centre [44], the Ipswich and East Suffolk MND clinic [45], and the Norfolk MND care and research network [46]. NICE produced the motor neuron disease: assessment and management [6] and the quality standard QS216 [47], while the National Neurosciences Advisory Group, in collaboration with MND specialists from the Association of British Neurologists (ABN), created the Optimal care pathway for MND [48]. There is a lack of unified UK-wide guidance on symptom management, for example, 3 different health boards produced nutrition and gastrostomy management guidance [42,43,46] and 3 different health boards produced respiratory management guidance [42,43,49]. The Motor Neurone Disease Association (MND Association) is an England-based registered charity and is prominent in supporting pwMND and their carers/ family members, HCPs, and researchers. The charity has also produced guidance documents and pathways such as the management of dysarthria [50] and respiratory impairment [51], and an audit tool to support access to a wheelchair [52]. In relation to the NAHP workforce, the MND Association developed a competency framework called the Allied Health Professionals' competency framework for progressive neurological conditions with content specific to multiple sclerosis, Parkinson's disease, and motor neuron disease [53]. This framework aimed to promote consistency of the roles and responsibilities of AHPs for people with progressive neurological conditions, maintain quality, safe, and effective care, and support professional development and revalidation for the recruitment and retention of specialist roles [53]. Within the framework are sections directed to dieticians, occupational therapists, physiotherapists, and speech and language therapists, and it is divided by neurological condition, band level, and the competencies required within the levels. It addresses the competencies required for all AHPs to support pwMND; however, it is not recognised as a standardised framework for AHPs across the UK.

Our findings suggested that some members of the MDT are unsure of the responsibilities and services that other members in the team provide; one respondent commented that this has led to "high and unrealistic expectations". Although the results revealed that 87.18% of the MND NAHP workforce are competent in palliative care, a study on the provision of palliative care in Northern Ireland found that carers perceived a lack of knowledge on holistic care needs from generalist palliative HCPs such as general practitioners, district nurses, and physiotherapists [54]. HCPs should provide specialised and evidence-based clinical information, because information that lacked evidence and was of poor quality only increased the difficulty for patients of accepting their diagnosis, which hinders the ability of the MDT to progress and make appropriate decisions [5]. In relation to the aspects of care in which the MND NAHP workforce engage, fewer than 62% of respondents engage in Respiratory Assessment, Muscle Management, and Specialist Medication Management (including the use of botox and/ or non-medical prescriber) and this is inconsistent with the recommendations from the NICE [6] guidelines. This indicated limitations in the conceptualisation and defined competencies for clinical professionals. Salary and job satisfaction was found to have a positive effect on the improvement of nurses' competence [55], and therefore, providing adequate support and funding may reduce the lack of competency and training in areas such as Respiratory Assessment,

Muscle Management, and Specialist Medication Management. Overall, the lack of tangible and informational support for HCPs can lead to patients becoming dissatisfied with their care and losing trust in their clinical care team [14,56].

With a number of general and specific guidelines produced across the UK, there is a need for a unified and standardised guidance that details the specific responsibilities and competencies that are essential in the MND NAHP workforce to maintain consistency and promote effective and high-quality care. Furthermore, the Scottish Intercollegiate Guidelines Network (SIGN), the Scottish equivalent of NICE, has not developed guidance or informed recommendations on the management of MND; hence, MND NAHPs in Scotland can only use the NICE [6] guidelines for evidence-based recommendations in clinical practice.

**Theme three: Benefits and challenges in providing direct care**

**Perceived benefits.** The findings revealed that the perceived benefits came from being valued by the patients and their carers/ family members and the core and the extended MDT, and also feeling satisfied about their work. This can be seen in the Table in S6 Table, where more than 80% of the workforce have a low risk of burnout. This was not what the authors hypothesised, considering how it has been documented in the literature that nurses are prone to burnout due to high workload [57]. It could be argued that the perceived benefits identified were inter-related. For example, it was previously found that there were lower burnout rates in nurses who considered themselves to have a "good working team", even though their workload was increasing [57]. Hence, lower burnout rates can be found in nurses who perceive themselves to be working in a positive working environment. This can be seen in the high percentage of respondents who answered that they always or often feel satisfied and valued in the core MDT and extended MDT. Furthermore, it was found that HCPs who provided MND care valued teamwork, as this involves information sharing and monitoring patients' effectively to provide a coordinated and timely response when the patients' needs change [58]. It was found that patients feeling satisfied with their care stems from the sympathy and engagement they receive from HCPs, and also from seeing them again overtime, which demonstrates that continuity of care is important [59]. The consequences of burnout are decreased in high job satisfaction [60] and this aligns with the scores in this report where the majority of the workforce have a low risk of burnout and a high percentage of "always" or "often" feeling satisfied with their work.

**Perceived challenges.** The perceived challenges that were found involved isolation, lack of direct funding, a high caseload with complex needs, and not enough time to provide quality care. This was associated with 14% and 12% being at medium and high risk of burnout, respectively. Isolation of HCPs and specifically emergency care nurses was found to be associated with a lack of interprofessional communication, professional support, and mentorship [61]. It was found that HCPs attributed health care funding to be most common obstacle that affected the delivery of specialist services [58]. This caused frustration due to feelings of being out of control and having limited options to improve patient care [58]. These findings are consistent with the current report, where respondents reported that a lack of direct funding was a barrier to providing care for pwMND. The allocation of resources is an often-discussed topic in the UK and there are health boards who support patients who live in deprived and rural areas who may have limited access to specialist services. However, analysis of the distribution of resources by the NHS from 2001 to 2011 revealed that investment in poorer or deprived areas increased survival and narrowed health inequalities per additional £1.00, compared to investment in affluent areas [62]. Understanding these organisational barriers within MND care can address the issues of funding and the lack of timely services. In relation to burnout, Taranu *et al.* [63] found that factors that were associated with burnout in HCPs were age, profession, workplace seniority, relationship status, and the presence of persons in care. Those with a higher level of work burnout were found to also have higher levels of quiet quitting, which is negatively associated with job satisfaction [64]. Quiet quitting describes employees who have a limited and lowered level of commitment towards their work activities, and therefore, do not feel the need to work above and beyond [65]. An example of this phenomenon was the COVID-19 pandemic where frontline

healthcare workers were experiencing high levels of burnout due to increased demands at work [66]. Consequently, it was found that the highest proportion of quiet quitting was found in nurses (67.4%) and physicians (53.8%) compared to other HCPs (40.3%) [64]. It was found that nurses who were quiet quitting also intended to resign, which shows that quiet quitting is only a temporary solution for nurses until they can find a better working environment [67]. Providing direct care for pwMND is challenging but it requires full commitment from each member of the MDT. Therefore, support for MND NAHPs is essential to reduce the risk of burnout.

Our results showed that 26% and 24% of the respondents were either at medium or high risk of exhaustion and emotional impairment, respectively. As emotional exhaustion is a factor that contribute to burnout [30], occupational stresses such as relationships at work, increased workload, and caring for patients were all negatively associated with emotional exhaustion [68]. The emotional and psychological characteristics of pwMND and their carers/ family members are relatively well-documented but the wellbeing of HCPs is not fully explored [12]. This demonstrates that the findings of this study are novel. Although the majority of respondents are not at risk of burnout, further investigation of the emotional and psychological wellbeing of the entire UK MND NAHP workforce and medical HCPs could provide new insights and inform resources for support.

## Implications and recommendations

The findings from this report have implications for policy, practice, and research in MND care. The discrepancies between the workforce and the adherence to national guidelines, vacancies within the MDT, and challenges in competencies should prompt targeted action to allocate additional resources and reduce these gaps. The authors recommend that a national competency programme or a Masters level course in MND care should be developed in cooperation with leaders of the workforce, policymakers, and charities to ensure standardisation of the training and competencies required to provide effective MND care across the UK. This could enhance adherence to national guidelines and address the workforce shortage and challenges in providing direct care. Furthermore, this report has identified that burnout and job satisfaction are crucial determinants of the wellbeing of HCP. Future research should develop a strategy to evaluate the entire workforce longitudinally to gain a better understanding how the time of experience affects HCP wellbeing, and to generalise the findings. Additionally, there is a need to explore organisational barriers and burnout preventative strategies to maintain a resilient and effective workforce.

## Strengths and limitations

This report had five strengths. Firstly, as this was the first census that evaluated the UK MND NAHP workforce, the employment of a mixed-methods approach allowed for greater depth and a holistic insight into the workforce. Secondly, it provided a comprehensive assessment which covered areas such as the changing dynamics of the MDT, the understanding of roles and responsibilities, and the perceived benefits and challenges of providing MND care. Thirdly, the descriptive analysis was performed by two of the authors to reinforce the robustness and validity of the study. Fourthly, this study was carried out with approval from NHS Lothian, and the respondents were anonymised to ensure confidentiality. Fifthly, this study utilised the Burnout Assessment Tool- Core dimensions (BAT-C) and -Secondary dimensions (BAT-S), which is a validated assessment tool that measures the risk of burnout among the workforce.

There were several weaknesses in this study that need to be addressed. As the survey was conducted after a single meeting, the level of transferability of the cross-sectional data will be low. For the same reason, and also due to the small sample size (n = 64), the conclusions drawn from this report may not be generalised to the entire workforce. Additionally, the use of qualitative data has potential for response bias. There were some respondents who had only recently started their post (e.g., 3 months prior to survey completion) and this would affect their burnout and job satisfaction levels and overall scores compared to those who have worked for longer.



## Conclusion

This report was able to provide novel and valuable insights into the UK MND NAHP workforce by evaluating aspects of their roles, competencies, experiences, and wellbeing. By using a mixed-methods approach, the study was able to answer the intended research questions and highlighted three important themes: the importance of a collaborative MDT to support the needs of patients, their carers/ family members, and HCPs themselves, the roles and competencies of the workforce, and the benefits and challenges of providing direct care. The challenges that this report identified included vacancies within the MDT, limited access to services, and organisational barriers that impacted the delivery of care. Despite these challenges, the majority of the workforce feels strongly satisfied with their job and feels valued by the MDT and their patients and carers/ family members. Furthermore, the workforce mostly adheres to national guidelines on MND care. However, it is imperative that future research aims to maintain consistency and quality of care through a standardised competency or training programme, address organisational barriers, and target interventions to support the workforce to enhance their wellbeing and resilience. Overall, the results have contributed to an understanding of the NAHPs who work in MND care serves as a starting point for further research, and prompts policy changes to improve care delivery to pwMND and their carers/ family members.

## Supporting information

**S1 Table. Summary table of participant demographics.** n, sample size; %, percentage.
(DOCX)

**S2 Table. Qualification Demographic. A. Qualification by Clinical Nurses (MND and non-MND specific).** MND, Motor Neuron Disease; n = sample size. **B. Qualifications by AHP.** AHP, Allied Health Professional; MN/MSc, Masters of Nursing/Masters of Sciences; BN/BSc, Bachelor of Nursing/Bachelor of Sciences; n, sample size; %, percentage.
(DOCX)

**S3 Table.** Employment Demographics. **A. Country of Work by Clinical and Non-clinical HCP.** HCP, Healthcare Professional; n = sample size; % = percentage. **B. Clinical vs Non-clinical and Full-time vs Part-time HCP.** HCP, Healthcare Professional; n = sample size; % = percentage. **C. Time of work per week (hours), Worked in MND care (months), Number of Patients (n), Time in Current Post (weeks).** MND, Motor Neuron Disease; n = sample size; % = percentage. **D. Vacancies in Core MDT**. MDT, Multidisciplinary Team; n = sample size; % = percentage.
(DOCX)

**S4 Table.** Clinical Scoring. **A. Competencies of MND care based on the NICE guidelines (2016) and NHS MND Advanced Clinical Nurse Specialist Pillars of Practice (2019).** MND, Motor Neuron Disease; NICE, National Institute for Health Care and Excellence; NHS, National Health Service; MDT, Multidisciplinary Team; n, sample size; %, percentage. **B. HCP/ Specialty Involvement in Core MDT**. HCP, Healthcare Professional; MDT, Multidisciplinary Team; n = sample size; % = percentage. **C. Established Relationship with NHS Services**. NHS, National Health Service; AAC, Augmented and Alternative Communication; n, sample size; %, percentage.
(DOCX)

**S5 Table.** Clinical Satisfaction Scores. MDT, Multidisciplinary Team; n, sample size; %, percentage.
(DOCX)

**S6 Table. Burnout Assessment Tool.** The clinical cut-off scores are grouped into three categories: 1) "Without Burnout"-Green, is between 1.00 to 2.58, 2) "At-Risk for Burnout"-Amber, the score is between 2.59 to 3.01, 3) "Likely to be Burned-Out"- Red, the score is between 3.02 to 5.00. A red score indicates that the specificity is greater than or equal to 0.90 that corresponds to a maximum misclassification of 10%. n = sample size; % = percentage. **A. Burnout Assessment**



Tool-Core (BAT-C) Total Scores. BAT-C, Burnout Assessment Tool-Core; n, sample size; %, percentage; N/A, Not Available. **B. Burnout Assessment Tool-Core (BAT-C) by Clinical HCP.** BAT-C, Burnout Assessment Tool-Core; HCP, Healthcare Professional; n, sample size; %, percentage; N/A, Not Available. **C. Burnout Assessment Tool-Core (BAT-C) by Non-clinical HCP**. BAT-C, Burnout Assessment Tool-Core; HCP, Healthcare Professional; n, sample size; %, percentage; N/A, Not Available. **D. Burnout Assessment Test-Secondary (BAT-S) Total Scores.** BAT-C, Burnout Assessment Tool-Core; n, sample size; %, percentage; N/A, Not Available. **E. Burnout Assessment Tool-Secondary (BAT-S) by Clinical HCP.** BAT-C, Burnout Assessment Tool-Core; HCP, Healthcare Professional; n, sample size; %, percentage; N/A, Not Available. **F. Burnout Assessment Tool-Secondary (BAT-S) by Non-Clinical HCP.** BAT-C, Burnout Assessment Tool-Core; HCP, Healthcare Professional; n, sample size; %, percentage; N/A, Not Available
(DOCX)

**S7 Table.** Burnout Assessment Tool questionnaire. A. Burnout Assessment Tool-Core (BAT-C) symptoms. B. Burnout Assessment Tool-Secondary (BAT-S) symptoms.
(DOCX)

## Acknowledgments

The team would like to thank all the MND NAHP Consortium who have contributed to this report and to all the four charities who supported the inaugural MND NAHP meeting.

This is the full membership list of the UK MND NAHP Consortium:

Kate Adams, Deborah Adair, Deborah Armstrong, Harriet Bailey, Christine Batts, Claire Bennett, Wendy Bennett, Sharon Bell, Andy Bethall, Lyn Bevan, Caroline Bidder, Suzanne Byrne, Julie Carson, Bernie Chapman, Theresa Chiwera, Gill Craig, Caroline Davis, Emma Davis, Victoria Edwards, Fiona Eldridge, Clare Erridge, Kate Ferguson, Moira Flett, Louise Gardiner, Pauline Gallagher, Andy Hamilton, Marie Hand, Janice Hatrick, Anthony Hanratty, Maggy Hevicon, Lynne Hills, Nadine Hodgson, Samantha Holden, Justine Hudson, Katherine Johns, Kim Kelly, Clare Lang, Helen Lennox, Erica Littleworth, Elizabeth MacDonald, Pauline McDonald, Laura Marshall, Joanna Matheson, Alison McEleney, Alison McBain, Kitty Millar, Rosemary Morris, Rebecca Picton, Beth Pudjinanto, Bally Purewal, Amelia Roberts, Nicola Ryder, Gowri Saravanan, Julie Scott, Phillipa Sharpe, Jean Sowerby, Tiffany Stewart, Susan Stewart, Gill Stott, Rhea Sutcliffe, David Thomson, Rachel Thomson, Rebecca Thurman, Katarzyna Tluchowska, Emma Townsley, Karen Twist, Naomi Unsworth, Ruth Valentine, Michaela Waltho, Carolyn Webber, Claire White, Hayley Williams, Sarah Woolhead, Stacy Young

The lead author for this group is Judith Newton (judith.newton@ed.ac.uk).

## Author contributions

**Conceptualization:** Ethan Stoker.

**Data curation:** Isaac Chau, Ethan Stoker.

**Formal analysis:** Isaac Chau, Ethan Stoker, Jessica Gill.

**Funding acquisition:** Judith Newton.

**Investigation:** Ethan Stoker.

**Methodology:** Ethan Stoker.

**Project administration:** Isaac Chau.

**Supervision:** Judith Newton.

**Writing – original draft:** Isaac Chau, Ethan Stoker.

**Writing – review & editing:** Isaac Chau, Judith Newton.



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
