## [Decision Letter · Decision Letter 0]

PONE-D-25-05908An evaluation of the United Kingdom Motor Neuron Disease Nurses and Allied Health Professionals workforce: A CensusPLOS ONE

Dear Dr. Chau,

Thank you for submitting your manuscript to PLOS ONE. After careful consideration, we feel that it has merit but does not fully meet PLOS ONE’s publication criteria as it currently stands. Therefore, we invite you to submit a revised version of the manuscript that addresses the points raised during the review process.

We look forward to receiving your revised manuscript.

Kind regards,

Masoud Rahmati

Academic Editor

PLOS ONE

2. One of the noted authors is a consortium [The United Kingdom Motor Neuron Disease Nurses and Allied Health Professionals Consortium]. In addition to naming the author group, please list the individual authors and affiliations within this group in the acknowledgments section of your manuscript. Please also indicate clearly a lead author for this group along with a contact email address

Reviewers' comments:

Reviewer's Responses to Questions

**Comments to the Author**

1. Is the manuscript technically sound, and do the data support the conclusions?

Reviewer #1: Yes

2. Has the statistical analysis been performed appropriately and rigorously? 

Reviewer #1: I Don't Know

3. Have the authors made all data underlying the findings in their manuscript fully available?

Reviewer #1: Yes

4. Is the manuscript presented in an intelligible fashion and written in standard English?

Reviewer #1: Yes

5. Review Comments to the Author

Reviewer #1: This research is very helpful in assessing the provision of MND care in the UK quantitatively and qualitatively.

## abstract

I would suggest to move the proportions at medium and high risk of burnout to the results section.

## Methods

Please elaborate on the uniqueness of the research setting when justifying the mixed methods methodology. In what regards is it unique and not comparable to multidisciplinary clinics for other neurodegenerative diesases?

Regarding compliance with national frameworks, were participants asked about their service in general or their individual scope of work? (last paragraph of page 5)? The last sentence on page seems to lack a noun: "..the relevant aspect of care with which (?they) where involved...."

Was the questionnaire piloted? If possible please elaborate on the development of the questionnaire.

"For the work experience and wellbeing analysis, only data from the

clinical professionals was used in order to ensure validity and relevance of the themes developed."

Which participants were excluded on this basis?

Please spell out the abbreviation "AAHP". Is the Association of Afghan Health Professionals?

"As the majority of data analysis was performed between

categorical data sets and was normally distributed, the main test used was the chi squared test. For

the continuous data sets (such as years of experience), one-way ANOVA tests were used."

Please specifically mention which test (ANOVA, chi squared test, t-test) was applied to each set of data and include the results of these tests.

In the spirit of mixed methods research I suggest to include the most important tables in the main text rather than the appendix, for example the demographics, highest training level achieved and job title.

6. PLOS authors have the option to publish the peer review history of their article (what does this mean? ). If published, this will include your full peer review and any attached files.

**Do you want your identity to be public for this peer review?** For information about this choice, including consent withdrawal, please see our Privacy Policy .

Reviewer #1: No

---

## [Author Response · Author response to Decision Letter 1]

16 Jun 2025

Dear Dr Rahmati,

Firstly, thank you so much for taking the time to peer-review this manuscript and inviting us to submit a revised version. This letter serves to respond to each point raised by yourself as academic editor, and reviewer 1.

To address each point made by the academic editor:

1. I have changed the formatting of the manuscript to match the formatting guidelines of PLOS ONE. This includes the file name, the Abstract section, the section headings, the in-text citations, titles, captions, and legends of the supporting information, and the referencing style. To align with PLOS ONE referencing guidelines, the in-text citations and Reference section has been changed to Vancouver referencing style. I have removed the title page from the manuscript because it should be in a separate document.

2. I have included the full membership list of the consortium, [The United Kingdom Motor Neuron Disease Nurses and Allied Health Professionals Consortium], to the Acknowledgements section of the manuscript and I have clearly indicated that Judith Newton is the lead author for this group along with her contact email address. I have not indicated the affiliations for each consortia author as the PLOS ONE Title, Author, Affiliations formatting guidelines have stated, Consortia/Group authors can have affiliations but it is not required.

3. I have changed the captions for supporting info to align with the PLOS ONE Supporting information guidelines where item names contain an “S” and number in order of appearance. Separate tables combined into one have been renamed with an “S”, a whole number, and a letter according to the order of appearance.

4. I have reviewed the reference list to ensure that it is complete and correct. I have noted in the revised manuscript with tracked changes the references that were removed or replaced with relevant current references.

To address each point made by reviewer 1:

Abstract: I would suggest to move the proportions at medium and high risk of burnout to the results section.

I have moved the proportions at medium and high risk of burnout to the results section of the abstract.

Methods: Please elaborate on the uniqueness of the research setting when justifying the mixed methods methodology. In what regards is it unique and not comparable to multidisciplinary clinics for other neurodegenerative diesases?

I have elaborated on the uniqueness of the research setting as this study captured the attendees of the inaugural United Kingdom motor neuron disease nurses and allied health professionals gathering, which allowed for a unique setting for data collection compared to a multidisciplinary clinic.

Regarding compliance with national frameworks, were participants asked about their service in general or their individual scope of work? (last paragraph of page 5)? The last sentence on page seems to lack a noun: "..the relevant aspect of care with which (?they) where involved...."

Regarding compliance with national frameworks, the participants were asked about their service in general. and I have included the missing word “they” in the indicated sentence.

Was the questionnaire piloted? If possible please elaborate on the development of the questionnaire.

The questionnaire was piloted with the members of staff in the Anne Rowling Regenerative Neurology Clinic. This information has now been added in Line 183-184 of Page 5. Two clinical and two non-clinical members of staff from the Anne Rowing clinic piloted the survey and they were relevant to this work within the area of motor neuron disease research.

"For the work experience and wellbeing analysis, only data from the clinical professionals was used in order to ensure validity and relevance of the themes developed." Which participants were excluded on this basis?

Participants who were non-clinical healthcare professionals and who were not a nurse or allied health professionals were excluded from the analysis. This information has now been added in Line 230-234 of Page 6.

Please spell out the abbreviation "AAHP". Is the Association of Afghan Health Professionals?

I have clarified and spelled out the abbreviation “AAHP”, which was in fact a typo as it should have been “AHP” and it stands for “Allied Health Professional”. This has been corrected in the revised manuscript.

"As the majority of data analysis was performed between categorical data sets and was normally distributed, the main test used was the chi squared test. For the continuous data sets (such as years of experience), one-way ANOVA tests were used." Please specifically mention which test (ANOVA, chi squared test, t-test) was applied to each set of data and include the results of these tests.

Within the data analysis section, I have clarified which statistical method was used for which data set and I have included a paragraph in the results section in Line 251-257 of Page 7 that the initial one-way ANOVA test and Chi-squared test were not statistically significant across the main variables, and for this reason, I have justified why the core authors decided to only use descriptive statistics.

In the spirit of mixed methods research I suggest to include the most important tables in the main text rather than the appendix, for example the demographics, highest training level achieved and job title.

Thank you for the suggestion. In the spirit of mixed-methods research, I have included a summary table of the important participant demographics within the main text rather than in the supporting information section, which includes sex distribution, age ranges, ethnicity, highest qualification achieved, and job title.

We believe that the manuscript has been significantly improved by these revisions, and hope that it is now suitable for publication in PLOS ONE.

Yours sincerely,

Isaac Chau*

*Corresponding author

---

## [Decision Letter · Decision Letter 1]

An evaluation of the United Kingdom Motor Neuron Disease Nurses and Allied Health Professionals workforce: A Census

PONE-D-25-05908R1

Dear Dr. Chau,

We’re pleased to inform you that your manuscript has been judged scientifically suitable for publication and will be formally accepted for publication once it meets all outstanding technical requirements.

Kind regards,

Masoud Rahmati

Academic Editor

PLOS ONE

Additional Editor Comments (optional):

Reviewers' comments:

Reviewer's Responses to Questions

**Comments to the Author**

1. If the authors have adequately addressed your comments raised in a previous round of review and you feel that this manuscript is now acceptable for publication, you may indicate that here to bypass the “Comments to the Author” section, enter your conflict of interest statement in the “Confidential to Editor” section, and submit your "Accept" recommendation.

Reviewer #1: All comments have been addressed

2. Is the manuscript technically sound, and do the data support the conclusions?

Reviewer #1: Yes

3. Has the statistical analysis been performed appropriately and rigorously? 

Reviewer #1: Yes

4. Have the authors made all data underlying the findings in their manuscript fully available?

Reviewer #1: Yes

5. Is the manuscript presented in an intelligible fashion and written in standard English?

Reviewer #1: Yes

6. Review Comments to the Author

Reviewer #1: (No Response)

7. PLOS authors have the option to publish the peer review history of their article (what does this mean? ). If published, this will include your full peer review and any attached files.

**Do you want your identity to be public for this peer review?** For information about this choice, including consent withdrawal, please see our Privacy Policy .

Reviewer #1: No

---

## [Editor Report · Acceptance letter]

PONE-D-25-05908R1

PLOS ONE

Dear Dr. Chau,

I'm pleased to inform you that your manuscript has been deemed suitable for publication in PLOS ONE. Congratulations! Your manuscript is now being handed over to our production team.

Kind regards,

on behalf of

Dr. Masoud Rahmati

Academic Editor

PLOS ONE